# Effect of Rod-like Structure on Fatigue Life, Short Surface Crack Initiation and Growth Characteristics of Extruded Aluminum Alloy A2024 (Analysis via Modified Linear Elastic Fracture Mechanics)

**DOI:** 10.3390/ma14247538

**Published:** 2021-12-08

**Authors:** Kenichi Masuda, Sotomi Ishihara, Hiroshi Shibata, Noriyasu Oguma

**Affiliations:** 1Department of Mechanical Engineering, University of Toyama, Gofuku 3190, Toyama 930-8555, Japan; sotomi.ishihara@gmail.com (S.I.); oguma@eng.u-toyama.ac.jp (N.O.); 2National Institute of Technology, Toyama College, Toyama 939-8630, Japan; shibata@nc-toyama.ac.jp

**Keywords:** extruded aluminum alloy, rod-like structure, fatigue crack initiation, fatigue crack growth, modified linear elastic fracture mechanics

## Abstract

In the Al alloy A2024-T3 extruded material, a rod-like structure is generated parallel to the extrusion direction. In this study, the effects of rod-like structures on fatigue crack initiation and growth behavior were comprehensively investigated. Two types of specimens were used in a fatigue experiment, in which the direction of the load stress amplitude was parallel (specimen P) and perpendicular (specimen V) to the rod-like structure. Based on the experimental and analytical results, the following findings were obtained regarding the fatigue life, location of crack initiation, and fatigue crack growth behavior. Because the fatigue life of specimen P was longer than that of specimen V, it is inferred that the rod-like structure significantly affects the fatigue life. In specimen P, fatigue cracks were generated from the grain boundaries of the Al matrix. By contrast, in specimen V, cracks were generated from the Cu–Mg-based intermetallic compound in the Al matrix. In specimen P, fatigue cracks were more likely to propagate across the rod-like structure, which decreased the fatigue crack growth rate. In specimen V, fatigue cracks did not propagate across the rod-like structure; instead, they propagated through the Al matrix. Therefore, the fatigue crack growth resistance of specimen V was lower than that of specimen P. The relationship between the fatigue crack growth rate and the modified linear elastic fracture mechanics parameter could be used to predict the S–N curve (stress amplitude vs. fatigue life) and fatigue crack growth behavior. The predicted results agreed well with the experimental results.

## 1. Introduction

Al alloys are widely used as structural members to reduce the weight of machines and structures. If a structural member is repeatedly stressed, cracks may appear from the deteriorated section, which could eventually result in its destruction. Therefore, to design machines and structures with high fatigue resistance, it is necessary to clearly understand the initiation and growth behavior of fatigue cracks as well as the fatigue life (N_f_) more accurately.

The fatigue crack growth (hereafter referred to as FCG) properties of Al alloys, with respect to corrosive environments, have been investigated extensively. For example, Stanzl–Tschegg et al. [1] used Al alloy A2024-T3 to study FCG behavior in a very high cycle fatigue region in moist, dry, and vacuum environments at a cyclic speed of 20 kHz. They reported an environmental effect on the threshold level of FCG (ΔK_th_) and the fatigue fracture surface. Meanwhile, van der Walde et al. [2] investigated the fatigue behavior of pre-corroded A2024-T3 plate specimens. They reported the generation and growth characteristics of multiple cracks that developed from corrosion pits. Menan et al. [3] investigated the interaction among forces, environments, and structures on the FCG properties of A2024 in corrosive environments.

Additionally, the FCG characteristics of actual structural members have been investigated. Liljedahl et al. [4] investigated the interaction between residual stress and FCG velocity using welded A2024-T351. They reported that the residual stress must be evaluated accurately to understand the FCG velocity of the welded material. Nejad et al. [5] investigated FCG behavior and estimated the N_f_ of the rivet joint of an A2024 plate both numerically and experimentally. Samaei et al. [6] investigated the effect of tightening torque (clamping force) on the FCG velocity and stress intensity factor K of cracked simple bolts and hybrid joints.

The FCG properties of short cracks in Al alloys have been investigated extensively. For example, HaIIiday et al. [7] conducted a four-point bending fatigue experiment using an A2024-T351 alloy to investigate the interaction between short cracks with a length of approximately 20 μm and crystal grains. They reported that fatigue cracks initiated from fine constituent particles in the material, and that the FCG velocity of the short cracks was higher than that of long through-thickness cracks. In addition, Zhang et al. [8] performed in-situ observations via scanning electron microscopy (SEM) using an aluminum alloy composed of fine crystals. They reported that short cracks as long as 2 μm grew along the shear band. Donnelly et al. [9] investigated the FCG properties of short cracks with lengths ranging from 10 μm to 0.5 mm using the A7705-T6 alloy. They investigated the differences in FCG properties obtained from plane and rotating bending fatigue experiments. lshihara et al. [10] investigated the FCG properties of short cracks with a length of 20 μm or more from the fatigue process of cast aluminum alloy W319-T7. These properties were analyzed using the modified linear elastic fracture mechanics (MLEFM) parameter, M [11,12,13,14]. They reported the effectiveness of using M to analyze short fatigue crack growth behavior.

A2024-T3 extruded products have been widely used in aircraft and automobiles. Recently, the authors observed the presence of a rod-like structure in the A2024-T3 extruded material (A2024-T3). However, the effect of the rod-like structure in A2024-T3 on fatigue behavior has yet to be elucidated.

The purpose of this study was to investigate the effects of rod-like structures generated in A2024-T3 on the N_f_, fatigue crack initiation (FCI) behavior, and FCG behavior. Therefore, in a fatigue experiment, we used two types of specimens, one in which the direction of the load stress amplitude σ_a_ was parallel to the rod-like structure (specimen P) and the other in which the direction was perpendicular to the rod-like structure (specimen V). During the fatigue process, the specimen surface was continuously observed using the replica method. The initiation and growth characteristics of micron-sized cracks were investigated using atomic force microscopy (AFM). Furthermore, the Nf and FCG behavior were analyzed based on the MLEFM parameter M [11,12,13,14]. The analysis results agreed well with the experimental results.

## 2. Microscopic Structure of Test Material

A commercially available A2024-T3 extruded material (round bar, diameter 15 mm) was used as the test material. Its chemical compositions are listed in Table 1. As shown, Cu, Mg, Mn, Zn, and Fe were added to the base material Al.

Tensile specimens were prepared using a round bar in the extrusion direction, and tensile experiments were subsequently conducted. The results are presented in Table 2, which shows that the yield strength and tensile strength were higher than those of the typical A2024.

### 2.1. Identification of Microscopic Structure

After the material surface was polished mechanically, it was etched with a solution containing hydrochloric acid (8 mL), nitric acid (2 mL), hydrogen fluoride (2 mL), and water (8 mL). Figure 1a shows the optical micrographs of the A2024-T3 extruded material in the longitudinal and cross sections after etching. As shown in the figures, black stripes were observed parallel to the extrusion direction in the longitudinal section (left side). In the cross section (right side), black structures of various shapes, including a round shape, were observed. Therefore, in the A2024-T3 extruded material, a rod-like structure was generated in the extruded direction. It was presumed that this rod-like structure was generated by the plastic flow of the material during extrusion. A similar structure has been observed in extruded magnesium alloy [15,16,17].

To observe the rod-like structure more clearly, a more detailed photograph of the microscopic structure in the longitudinal section, after etching obtained via optical micrography, is shown in Figure 1b. As presented in the figure, a light-blue rod-like structure (black arrow in the figure) with a width (diameter) of 20–70 μm appeared parallel to the extrusion direction.

A schematic diagram of the microstructure is shown on the left side of Figure 1b. The microscopic tissues observed can be classified into four phases: A, B, C, and D. Phase A is the region with a light blue stripe (rod-like structure); phase B, the black region in the rod-like structure; phase C, the region with a pale white structure; and phase D, the region with a spherical inclusion inside phase C.

Phases A, B, C, and D were identified using an electron probe microanalyzer (SHIMADZU, EPMA-1500, Kyoto, Japan) and an X-ray diffractometer (Rigaku; RINT2200/PC/K, Tokyo, Japan). The result shows that phase A (rod-like structure) contained an Al–Cu–Mn intermetallic compound; phase B contained a locally segregated Al–Cu–Mn intermetallic compound within phase A; phase C contained the Al matrix; and phase D contained a Cu–Mg-based intermetallic compound in phase C.

### 2.2. Microhardness

The hardness of each phase was measured using a microhardness meter (Akashi, HM-102, Kanagawa, Japan). Hardness was measured at a load of 10 g and a holding time of 1 min. Microhardness was measured at seven to eight points in each phase, for a total of 30 points.

Table 3 summarizes the identification results of phases A, B, C, and D and the microhardness of each phase. As shown, the hardness (average measured values) of each phase was as follows: phase A (rod-like structure), HV = 182; phase B, HV = 207; phase C (Al base material), HV = 142; phase D, HV = 213. Therefore, the order of hardness from high to low was as follows: phase D (Cu–Mg intermetallic compound) ˃ phase B (black inclusions) ˃ phase A (rod-like structure) ˃ phase C (Al matrix).

### 2.3. Crystal Grain Size

The crystal grain size of the A2024-T3 extruded material was measured as follows: First, after etching the specimen surface, a replica film of the etched surface was obtained. This replica film was observed using AFM (SHIMADZU: Scanning Probe Microscope-9500 J2).

Figure 2 shows an example of the AFM results. As shown by the AFM image in Figure 2a, some grain boundaries can be observed due to the unevenness of the replica film. However, grain boundaries due to etching only were unclear. Therefore, the following method was used. The AFM needle was scanned along line segments A to B, as shown in Figure 2a. Figure 2b shows the relationship between the excitation voltage and the moving distance when scanning the needle. A voltage drop was observed at three locations (i.e., positions marked with **×**, **×**, and **×** in Figure 2a) and was assumed to represent grain boundaries on the line segments from A to B (Figure 2b). Therefore, these points were regarded as the grain boundaries.

The grain boundaries were determined based on both the abovementioned visual inspection of grain boundaries from AFM images and the decrease in the excitation voltage. The crystal grain size was determined using a linear-cutting method; the average crystal grain size obtained was 2.3 μm.

To confirm the AFM image measurement results, the crystal grain size was measured by directly observing the specimen surface after electrolytic polishing via SEM (HITACHI; S-530, Kanagawa, Japan). The SEM measurement results agreed well with the AFM results. The static strength of this test material (Table 2) is relatively high compared with that of a typical A2024-T3 alloy owing to the presence of the rod-like structure and the fine crystal grain size.

## 3. Specimen and Experimental Methods

### 3.1. Specimen

Specimens P and V were used to investigate the effect of the rod-like structure on the FCI and FCG characteristics of short cracks during the fatigue process of the extruded A2024-T3 material.

Specimens P and V were prepared from the same round bar with a diameter of 15 mm. The shapes and dimensions of both specimens are shown in Figure 3a,b, respectively. On the right side of Figure 3, the geometrical relationship between the rod-like structure (gray area) and stress amplitude σ_a_ is shown schematically. As shown in the figure, in specimens P and V, σ_a_ was parallel and perpendicular to the rod-like structure, respectively. The white region was the Al matrix, and inclusions (Cu–Mg-based intermetallic compounds) were present in that region.

The processing of specimen P was easy because the longitudinal direction of the specimen coincided with the axial direction of the round bar. By contrast, specimen V was processed via wire cutting such that the longitudinal direction of the specimen was the diameter direction of the round bar (see the schematic diagram in Figure 3b). Therefore, the length in the longitudinal direction of specimen V was reduced (10 mm). To adjust the length of the specimen, SUS304 round bar grips (Figure 3b) were welded to both ends of specimen V. Before welding, an adhesive was used to bond the test piece to the grip portion. Subsequently, spot welding was performed at four locations so that the effect of welding on the specimen did not increase.

At this time, it was processed meticulously to prevent the axis of the specimen from shifting. Two specimens of the same material but different shapes (specimens P and V) were prepared, and their S–N curves were compared. It was confirmed from the S–N curves that there was no significant difference between the two specimens.

All specimens were mirror-like finished using emery paper and diamond paste prior to the fatigue experiment.

### 3.2. Experimental Methods

#### 3.2.1. S–N Curve

The fatigue test was performed using a cantilever-type rotating bending fatigue machine. Fatigue tests with a stress ratio R = −1 were conducted in a laboratory environment (humidity 68%, temperature 25–30 °C) under a stress repetition rate of 30 Hz to analyze the S–N curves (σ_a_-N_f_ relationship) of specimens P and V, where N_f_ is the number of cycles to failure of the specimen.

#### 3.2.2. FCI and FCG Experiments

##### Constant σ_a_ Experiment

The FCG behavior of short surface cracks initiated on a smooth specimen surface was investigated using the replica method [18]. A σ_a_-constant fatigue experiment was performed, and the fatigue experiment was interrupted at regular intervals. At each interruption, a replica of the specimen surface was obtained using an acetyl cellulose film. The procedure above was continued until the specimen failed. After the specimen failed, the FCI and FCG characteristics of the fatigue process were studied by tracing back from the last replica (just before the specimen failure) to the first replica (at the beginning of the experiment). The interaction between fatigue cracks and the microstructure was investigated. Optical microscopy and AFM were performed to observe the replicas.

##### Constant ΔK experiment

To analyze the interaction between the fatigue cracks and rod-like structures more comprehensively, FCG experiments were conducted using specimen P under a constant ΔK condition. Here, ΔK = K_max_ − K_min_, where ΔK is the stress intensity factor range, and K_max_ and K_min_ are the maximum and minimum stress intensity factors, respectively. In this study, it was assumed that the negative region of ΔK at R = −1 did not have a crack growth driving force, and that K_min_ = 0. The ΔK value for the surface cracks was calculated using the following equation:ΔK = Yσ_a_(πa)^1/2^,(1)
where Y is the correction coefficient. Assuming that the crack shape of the surface crack is a semicircle, Y is 0.73 [19]. The fatigue fracture surface of the specimen was observed at a stress amplitude of 250 MPa using SEM. A semi-circular surface crack with a length of 6.7 μm and a depth of 3.5 μm was observed at the origin of the fracture. It was also observed that the shape of the crack could be approximated by a semicircle up to a length of approximately 2 mm. It is considered that when the length of the crack is short, the influence of the stress gradient due to bending is unlikely to occur.

As shown in Figure 3, the specimen does not have a parallel section and has an arcuate notch. Therefore, the stress at the minimum diameter of the specimen was determined using the formula for the strength of materials. The computed value was multiplied by the stress concentration factor of 1.04, due to the notch in the arc, to obtain the stress amplitude σ_a_.

Constant ΔK experiments were performed at ΔK = 1.2 and 2.0 MPam^1/2^. Here, ΔK was adjusted by adjusting σ_a_ based on the half-length (a) elongation of the crack.

## 4. Experimental Results

### 4.1. S–N Curves

Figure 4 shows the S–N curves of specimens P and V. For comparison, the S–N curve (broken line in the figure) of the rolled Al alloy 2024T-3 reported by Newman [20] is included in the figure. The arrows in the figure indicate that the specimen is not broken, but the experiment has been stopped. As shown in the figure, the fatigue limits of specimens P and V (σ_a_ at N_f_ = 10^7^) were 190 and 140 MPa, respectively. Therefore, the fatigue limit value of specimen P was 50 MPa greater than that of specimen V. Furthermore, based on a comparison between specimen V and the rolled Al alloy, it was observed that the S–N curves of both were similar.

Results from the S–N curve indicate that the fatigue strength and fatigue limit of the A2024-T3 extruded material were affected by the rod-like structure. Similar results have been reported for magnesium alloys [21].

### 4.2. FCI and FCG Behavior of Short Surface Cracks

#### 4.2.1. Specimen P

Using specimen P, the specimen surface during the fatigue process was observed continuously (using the replica method) at σ_a_ = 200 MPa, which is slightly higher than the fatigue limit of 190 MPa. Additionally, the FCI and FCG behaviors of the short surface cracks were investigated. N_f_ was 5.7 × 10^6^ cycles, and AFM was used to observe short surface cracks with a length of 10 μm or less.

Figure 5a,b show the AFM images for N/N_f_ = 10% and 31% for specimen P, respectively, where N is the number of stress cycles. The arrangement and shape of the crystal grains near the short cracks are shown at the bottom of each image. The arrangement and shape of these grains were analyzed based on the AFM images and the excitation voltage distribution (as explained in Section 2.3).

At N/N_f_ = 10%, as shown in Figure 5a, a short surface crack with a length of 1.9 μm was observed between a and b in the figure. Based on the schematic diagram shown at the bottom of Figure 5a, this crack initiated at or near the grain boundaries of crystal grains 10 and 11. For N/N_f_ = 31%, as shown in Figure 5b, the schematic diagram at the bottom of the figure shows that fatigue cracks (length 1.9 μm) generated between a and b developed gradually through the grain boundaries up to c–d (length 3.4 μm).

The difference in N between Figure 5a,b was 1.2 × 10^6^ cycles. Therefore, the FCG velocity, da/dN, increased at an extremely low speed of approximately 1 × 10^−12^ (m/cycle), owing to the fine crystal grains.

#### 4.2.2. Specimen V

Using specimen V, the FCI and FCG behavior of short surface cracks were continuously observed at σ_a_ = 150 MPa, which is slightly higher than the fatigue limit of 140 MPa. Figure 6 shows an example of the results obtained from an AFM image of the replica film. Figure 6a,b correspond to N/N_f_ = 0% and 16%, respectively. Here, N_f_ was 1 × 10^5^ cycles. In the AFM image before the fatigue loading shown in Figure 6a (N/N_f_ = 0), inclusions (Cu–Mg-based intermetallic compounds) were observed, as indicated by the arrows. At N/N_f_ = 16%, as shown in Figure 6b, fatigue cracks that initiated from the inclusion (arrows in the figure, Cu–Mg-based intermetallic compounds) developed to a final length of 12 μm. The da/dN at this time was 3.75 × 10^−10^ (m/cycle). A similar result for FCI has been obtained for magnesium alloys [17].

### 4.3. FCG Behavior of Short Surface Cracks

Based on the continuous observation of the fatigue process, the relationship between 2a and N was investigated for specimen P. The FCG behavior was investigated based on three σ_a_ values, i.e., 200 MPa, which is slightly above the fatigue limit, to 300 MPa, which represents high stress.

Figure 7a,b show the relationships 2a–N and 2a–N/N_f_, respectively. Multiple surface cracks occurred and developed further in the unnotched smooth specimen. Indications No. 1 to No. 3 in the figure are shown to distinguish a plurality of cracks.

Next, the FCG behavior of specimen V was observed. Figure 8a,b show the relationship between 2a–N and 2a–N/N_f_ of specimen V, respectively. In specimen V, the length of the test section was short, which resulted in a difficult experiment. Therefore, the FCG behavior could only be measured with a σ_a_ value of 150 MPa.

As shown in Figure 7b, in specimen P, a 1–2 μm short crack in 2a was initiated at N/N_f_ = ~10%, and developed further.

In specimen V (Figure 8b), a 10-μm-long short crack was initiated at N/N_f_ = 16%. Therefore, it was inferred that the N_f_ was primarily the FCG lifetime, N_g_, of the short surface cracks, including in the unnotched smooth specimen. This result, which was not affected by specimens P and V, was similar to those reported previously [18].

Figure 9 shows the da/dN−ΔK relationships of specimens P and V. da/dN was calculated using Figure 7 and Figure 8, and ΔK was calculated using Equation (1) for surface cracks.

For specimen P, the da/dN−ΔK relationships for the three σ_a_ values (200, 250, and 300 MPa) are shown. Focusing on da/dN at a constant ΔK value, it was observed that da/dN increased with σ_a_. Linear elastic fracture mechanics (LEFM) cannot be applied to short cracks because of the dependence of σ_a_ in the da/dN−ΔK relationship for short cracks [14,22]. Hence, da/dN was analyzed using the MLEFM parameter M (as presented in Section 5.2).

The da/dN−ΔK relationship for σ_a_ = 150 MPa of specimen V was similar to that of σ_a_ = 300 MPa for specimen P.

Although the difference in σ_a_ between the two specimens was 150 MPa, the da/dN of specimen V was greater than that of specimen P. The difference between the two specimens in terms of the FCG behavior corresponded to the difference between the two specimens in terms of the S–N curve (Figure 4). A detailed analysis is presented in Section 5.1.

## 5. Discussion

### 5.1. Effect of Rod-Like Structure on FCG Behavior

Using specimen P, the FCG experiment was performed under constant conditions of ΔK = 2.0 MPa√m. The interaction between FCG and the rod-like structure was observed in detail using the replica method. After performing the FCG experiment under a constant ΔK condition, the specimen surface was etched to reveal a rod-like structure and an Al matrix.

Figure 10a shows the interaction between the rod-like structure and fatigue crack. As shown, the crack width was wider than the actual crack width, owing to the effect of the etching. The black striped pattern in the vertical direction in the figure is the rod-like structure, and the gray pattern is the Al matrix. Symbols “a” to “l” are included in the figure to measure the da/dN of each section (local position). The crack developed from the left to the lower right in the figure, and the tip of the crack was “l”. At this time, crack length was approximately 170 μm. The extrusion direction is shown in the figure. The crack did not grow in a direction perpendicular to the load direction (extrusion direction); instead, they grew locally in a diagonal direction. Therefore, the crack was projected on a plane orthogonal to the load direction in order to measure its length.

The local da/dN measured via the replica method can be associated with the microstructure shown in Figure 10a. Hence, the da/dN in each microstructure could be measured. The measurement results are presented in Figure 10b. The vertical and horizontal axes of the figure represent da/dN and a, respectively. The gray and white areas in the figure indicate the rod-like structure and Al matrix, respectively. These regions are associated with the microstructure shown in Figure 10a.

As shown in Figure 10b, although an FCG experiment with a constant ΔK was conducted, the da/dN value changed in each section. The da/dN (average 5 × 10^−10^ m/cycle) in the rod-like structure was approximately 10% of that in the Al matrix (average 5 × 10^−9^ m/cycle). The hardness of the rod-like structure (HV182) was greater than that of the matrix phase (HV142); therefore, the former was difficult to deform. As such, the rod-like structure was assumed to be an obstacle when the fatigue crack propagated across the rod-like structure.

Meanwhile, in specimen V, as shown in Figure 6, fatigue cracks were initiated from phase D (Cu–Mg-based intermetallic compound) inside the Al matrix. As shown in Figure 3b, the loading direction of σ_a_ was perpendicular to the rod-like structure. In this case, the fatigue crack propagated in the Al matrix, and the probability of traversing across the rod-like structure was low. Therefore, the FCG velocity of specimen V was higher than that of specimen P because FCG resistance due to the rod-like structure did not exist.

### 5.2. Estimating S–N Curves and FCG Behavior Using MLEFM Parameter M

As mentioned in Section 4.3 (Figure 9), LEFM cannot be applied to the FCG behavior of short surface fatigue cracks. Therefore, the FCG behavior of short surface fatigue cracks was analyzed using the MLEFM parameter M [11,12,13,14] proposed by McEvily.

For the analysis, the FCG rule presented in Equation (2) was used as the basic equation [11]. It is noteworthy that Equation (2) has been verified for long through-cracks that satisfy the small-yield condition.
(2)$$\frac{\mathrm{da}}{\mathrm{dN}}{=A(\Delta K}_{\mathrm{eff}}-{\Delta K}_{\mathrm{effth}}{)}^{2}$$

Here, A is a material constant; ΔK_eff_ (=K_max_ − K_op_) and ΔK_effth_ are the effective stress intensity factor range and its threshold value, respectively; and K_max_ and K_op_ are the stress intensity factors at the maximum and crack openings, respectively. When analyzing the FCG behavior of the short-surface cracks, Equation (2) must be modified to accommodate the following three aspects [11,12,13,14]:

(a) Elastoplastic behavior: For short cracks, the small yield condition (LEFM) is not satisfied because the plastic zone size at the crack tip is large compared with the crack length. Irwin [23] proposed using a modified crack length method to incorporate elastoplastic behavior into LEFM, as expressed in Equation (3), where a_mod_ is the modified crack length, a is half actual crack length, and R_pz_ is the crack tip plastic zone size. Using Dugdale’s equation [24] to calculate R_pz_, a_mod_ is expressed as shown by Equation (3), where σ_Y_ represents the yield strength of the material;
(3)$$\begin{array}{c} a_{\mathrm{mod}}=a+\frac{1}{2}R_{\mathrm{pz}}=aF\\ \mathrm{where},\;F=\frac{1}{2}\left\{\left(\sec(\frac{\pi }{2}\frac{\sigma_{a}}{\sigma_{Y}})+1\right)\right\} \end{array}$$

(b) Crack closure behavior: In short cracks, K_op_ increases from 0 to K_opmax_ as the crack growth amount λ increases. Here, K_opmax_ is the K_op_ value for long-penetrating cracks. The trend of K_op_ in short surface cracks can be approximated using the following exponential function [25,26]:(4)$$K_{\mathrm{op}}=(1-e^{-k\lambda }{)K}_{\mathrm{opmax}}$$

Here, k is a parameter indicating the rate of increase of K_op_, and its value depends on the material used. Meanwhile, λ represents the amount of crack growth and is expressed as λ = a − r_e_, where r_e_ is the initial crack length (see condition (c));

(c) Endurance limit–threshold relationship: According to Kitagawa–Takahashi et al. [27], the threshold of FCG in short cracks is defined by the fatigue limit σ_w_ instead of ΔK_effth_. To incorporate this condition, the material constant r_e_, as expressed in Equation (5), is introduced into Equation (2) [13,14].
(5)$$r_{e}=(\frac{{\Delta K}_{\mathrm{effth}}}{\sigma_{w}}{)}^{2}\left\{\frac{1}{2\pi F\left(1+\sqrt{2}Y+0.5Y^{2}\right)}\right\}$$

By modifying Equation (2) based on (a)–(c) presented above, an equation that yields the FCG velocity for short-surface cracks can be obtained, as shown in Equation (6).
(6)$$\frac{\mathrm{da}}{\mathrm{dN}}=A[(\sqrt{{2\pi r}_{e}F}+Y\sqrt{\pi \mathrm{aF}}{)\sigma }_{a}-{(1-e}^{-k\lambda }{)K}_{\mathrm{opmax}}-{\Delta K}_{\mathrm{effth}}{]}^{2}$$

The first, second, and third terms on the right-hand side of Equation (6) correspond to the threshold condition for short crack growth (condition (c)), elastoplastic behavior (condition (a)), and crack closure (condition (b)), respectively.

The following equation is obtained by expressing the inside of the parentheses on the right side of Equation (6) with the MLEFM parameter M.
(7)$$\frac{\mathrm{da}}{\mathrm{dN}}={\mathrm{AM}}^{2}$$

To analyze the FCG behavior of the short cracks using Equations (6) and (7), the material constants such as ΔK_effth_, K_opmax_, and k must be determined. In this study, these values were obtained from the results of a study by de Matos et al. [28] pertaining to the A2024-T3 alloy. Table 4 lists the values. As an approximation, it was assumed that the values for specimens P and V were equal. As shown by the S–N diagram in Figure 4, specimens P and V had different σ_w_ values. Therefore, the r_e_ values (Equation (5)) for both specimens were different.

Figure 11 shows a graph depicting the da/dN–M relationship. Figure 11a shows the results for specimen P for the above mentioned three σ_a_ values. The solid line in the figure will be described later herein.

No systematic difference was observed among the three σ_a_ values in the da/dN–M graph, although the experimental data varied. Therefore, it was concluded that for the parameter M, three modifications to LEFM ((a), (b), (c)) were successfully incorporated.

Figure 11b shows the da/dN–M relationship (σ_a_ = 150 MPa) for specimen V. Additionally, the da/dN–M relationship of specimen P (straight line in Figure 11a) is shown. For comparison, the da/dN–(ΔK_eff_–ΔK_effth_) relationship [29] obtained by Bao for the long penetrating crack of A2024 is shown in the figure. He conducted FCG experiments with specimen thicknesses B of 0.3 and 6.35 mm.

As shown in Figure 11b, at a constant M value, the da/dN of specimen V was greater than that of specimen P. This might be because specimen V was unaffected by the rod-like structure, unlike specimen P (Section 5.1).

The da/dN–M relationship (MLEFM) for the short surface cracks of specimen V was consistent with the da/dN–(ΔK_eff_–ΔK_effth_) relationship (LEFM) for long penetrating cracks investigated by Bao. Furthermore, based on the straight line (slope 2) of specimen P, the da/dN–M of specimen V and da/dN–(ΔK_eff_–ΔK_effth_) relationships can be approximated based on a straight line with a slope of 2. This is a reasonable theoretical result.

When the da/dN–M relationships shown in Figure 11 were approximated using Equation (7), Equations (8) and (9) were obtained for specimens P and V, respectively.
(8)$$\frac{\mathrm{da}}{\mathrm{dN}}=2\times 10^{-10}M^{2}\;\mathrm{for}\;\mathrm{specimen}\;P$$
(9)$$\frac{\mathrm{da}}{\mathrm{dN}}=4\times 10^{-9}M^{2}\;\mathrm{for}\;\mathrm{specimen}\;V$$

Meanwhile, N_g_ can be calculated by numerically integrating Equations (8) and (9) from r_e_ to the critical half-length a_f_ (2 mm) of the fatigue crack. Because a_f_ is the crack length in the rapid unstable fracture region, it may be 2–4 mm; however, in this study, we assumed a_f_ = 2 mm.

Generally, N_f_ is expressed by the sum of the crack initiation lifetimes N_i_ and N_g_ (N_f_ = N_i_ + N_g_). As shown in Figure 7 and Figure 8, fatigue cracks initiated early in the fatigue process (N_i_/N_f_ = 5–20%). Therefore, N_i_ was smaller than N_g_ and negligible. Subsequently, N_f_ was approximated to be equal to N_g_ and was calculated as follows:(10)$$N_{f}≅N_{g}=\int_{N_{i}}^{N_{f}}dN=\int_{r_{e}}^{a_{f}}\frac{\mathrm{da}}{{\mathrm{AM}}^{2}}$$

The two curves shown by the solid lines in Figure 12 are the S–N curves calculated for specimens P and V using Equation (10). As presented in the figure, the calculation results and experimental data indicated good agreement.

The 2a–N and 2a–N/N_f_ relationships can be calculated by substituting Equations (8) and (9) into Equation (11).
(11)$$a-r_{e}=\int_{r_{e}}^{a}\mathrm{da}=\int_{N_{i}}^{N}{\mathrm{AM}}^{2}\mathrm{dN}\approx \int_{0}^{N}{\mathrm{AM}}^{2}\mathrm{dN}$$

Figure 13 compares the results of the crack growth curve for specimen P calculated using Equation (11) with the experimental results. In addition, Figure 14 compares the results of the crack growth curve for specimen V calculated using Equation (11) with the experimental result. The curves shown by the solid and broken lines in Figure 13 and Figure 14 indicate the calculation results. As presented in the figure, the calculation and experimental results indicated good agreement. Hence, it can be concluded that the MLEFM parameter M is effective and useful for the analysis of the FCG behavior of short surface cracks and the S–N curve.

## 6. Conclusions

In this study, the effect of a rod-like structure on the fatigue and FCG behavior of an extruded A2024-T3 alloy was investigated. Rotating bending fatigue experiments were conducted using two types of specimens, i.e., specimens P and V. Using the replica method, the fatigue damage process was continuously observed to investigate the FCI and FCG behavior of both specimens. The results are summarized as follows:(1)In the extruded A2024-T3 alloy, a rod-like structure (Al–Cu–Mn-based intermetallic compound) was formed parallel to the extrusion direction;(2)The N_f_ of specimen P was longer than that of specimen V. The fatigue resistance of specimen P increased owing to the rod-like structure;(3)In specimen P, fatigue cracks were initiated from the grain boundaries of the Al matrix. By contrast, in specimen V, cracks were initiated from a Cu–Mg-based intermetallic compound with a high hardness in the Al matrix;(4)Based on the FCG experiment under constant ΔK conditions, in specimen P, the FCG velocity decreased when the fatigue crack passed through the rod-like structure. This is because the hardness of the rod-like structure (HV182) was greater than that of the Al matrix (HV142); hence, the former was difficult to deform and adversely affected the FCG behavior.(5)Using the da/dN–M relationship, the S–N curve and FCG behavior could be calculated. The calculated results agreed well with the experimental results.(6)By combining the replica method with a scanning atomic force microscope, fatigue cracks with a length of several microns were continuously observed, and it was determined that fatigue cracks occurred in the early stages of the fatigue process.

## Figures and Tables

**Figure 1 materials-14-07538-f001:**
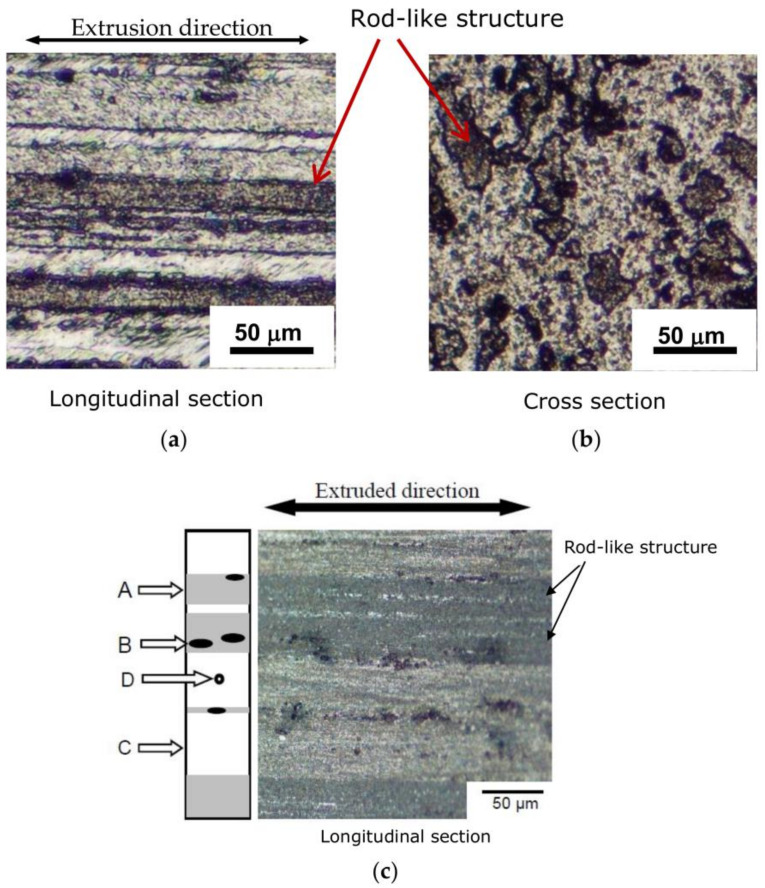
Extruded A2024-T3 microstructure. (**a**) Longitudinal section of test material. (**b**) Cross section of test material. (**c**) Analysis of microstructure.

**Figure 2 materials-14-07538-f002:**
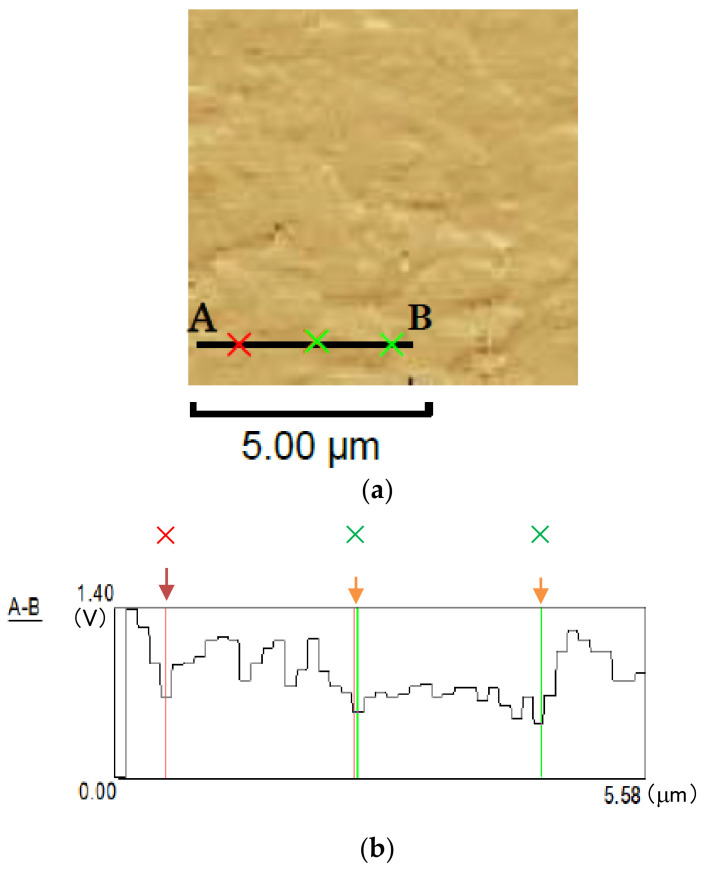
Determination of grain boundary position via AFM. (**a**) AFM image of replica film; (**b**) Change in excitation voltage (V) on line segment A–B. Three locations marked with **×**, **×**, and **×** in these figures correspond to grain boundaries existing on the line segment (from A to B).

**Figure 3 materials-14-07538-f003:**
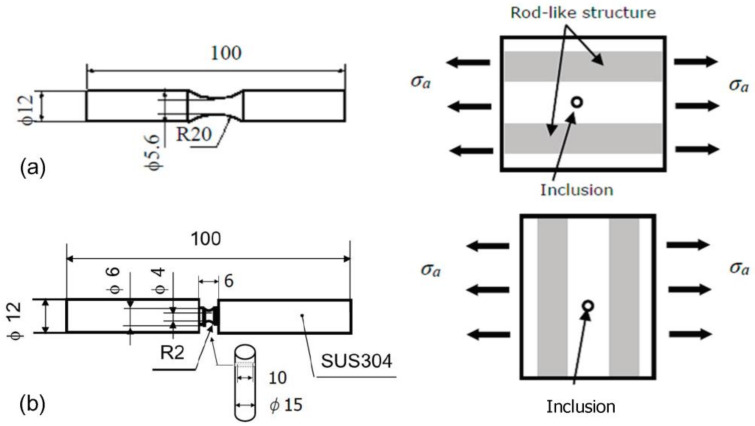
Shape and dimensions of fatigue specimens; relationship between σ_a_ and rod-like structure (schematic diagram). (**a**) Specimen P; (**b**) specimen V.

**Figure 4 materials-14-07538-f004:**
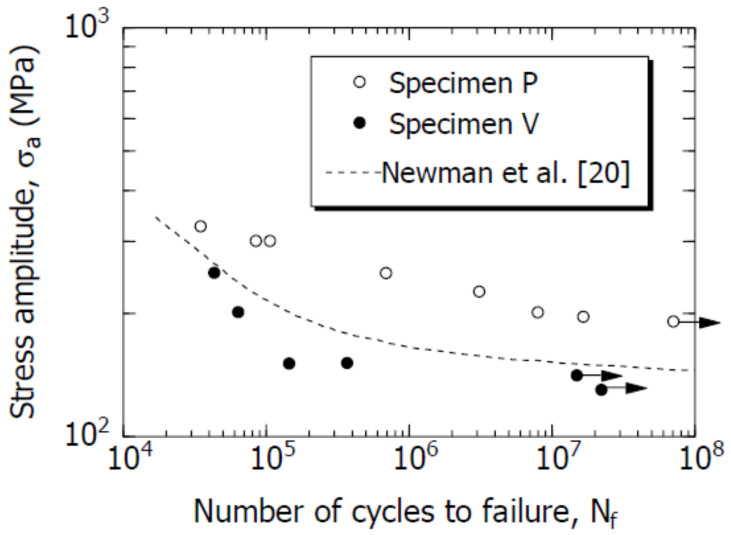
S–N curves of specimens P and V based on laboratory air.

**Figure 5 materials-14-07538-f005:**
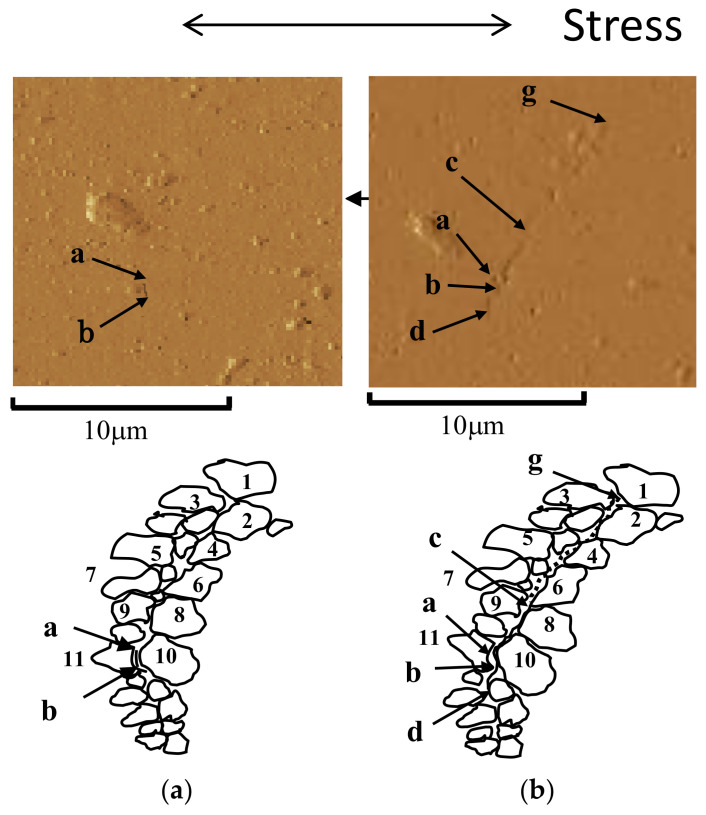
Interaction between cracks and microstructure (specimen P, σ_a_ = 200 MPa, N_f_ = 5.7 × 10^6^). (**a**) N/N_f_ = 10%; (**b**) N/N_f_ = 31%.

**Figure 6 materials-14-07538-f006:**
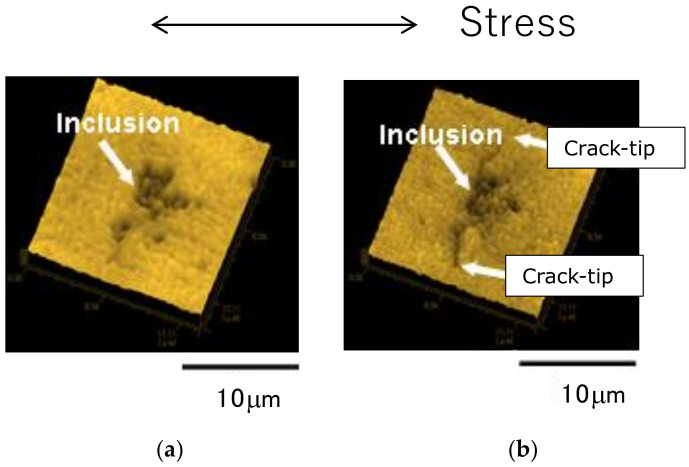
Continuous observation of specimen surface during fatigue process (specimen V, σ_a_ = 150 MPa, N_f_ = 1.0 × 10^5^ cycles). (**a**) N = 0 cycles (N/N_f_ = 0%); (**b**) N = 16,000 cycles (N/N_f_ = 16%).

**Figure 7 materials-14-07538-f007:**
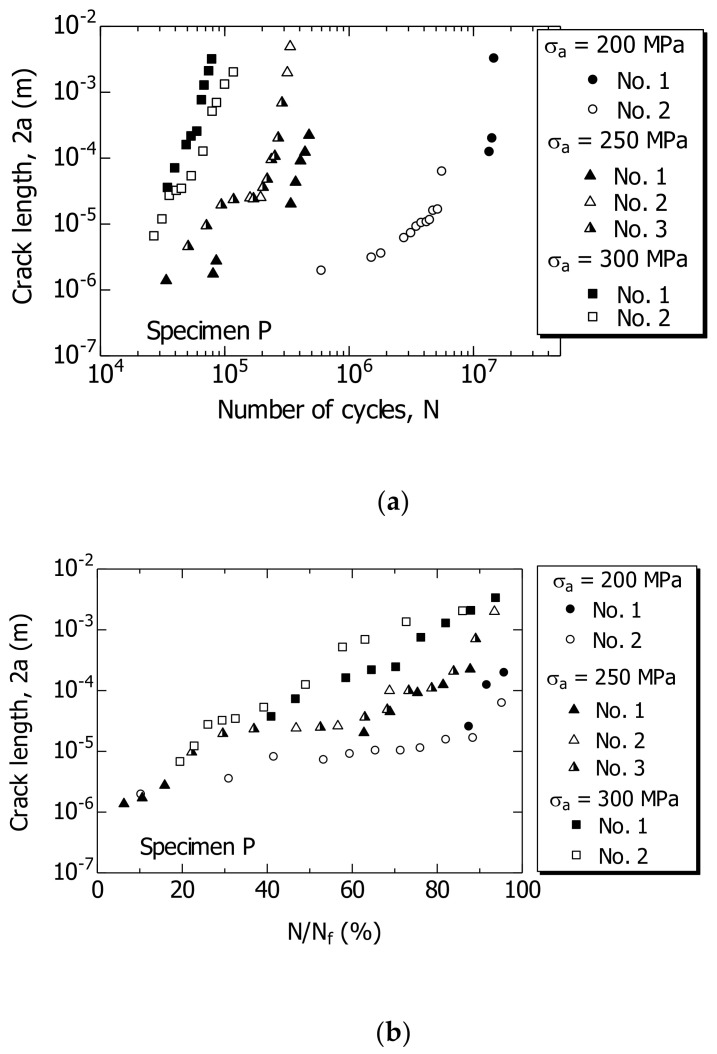
Changes in crack length 2a during fatigue process of specimen P. (**a**) 2a–N; (**b**) 2a–N/N_f_.

**Figure 8 materials-14-07538-f008:**
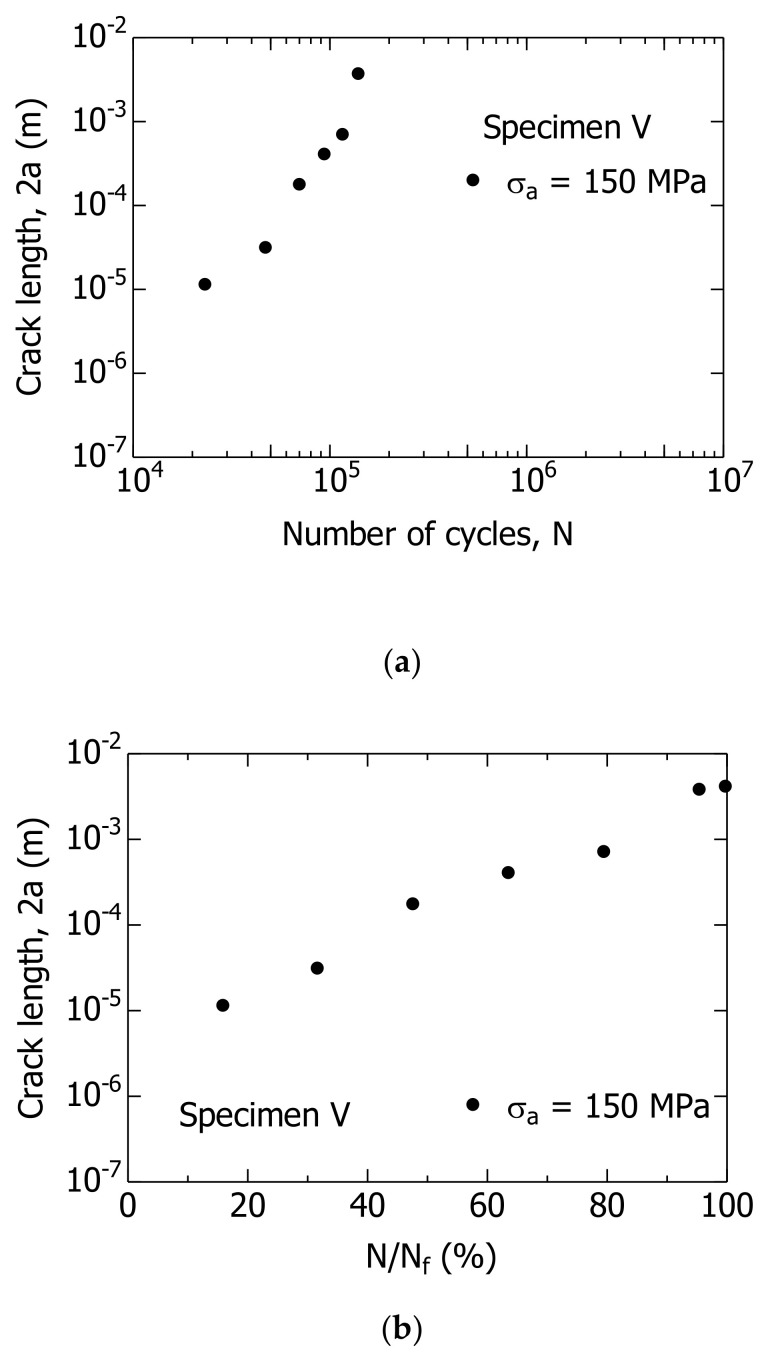
Changes in crack length 2a during fatigue process of specimen V. (**a**) 2a−N; (**b**) 2a−N/N_f_.

**Figure 9 materials-14-07538-f009:**
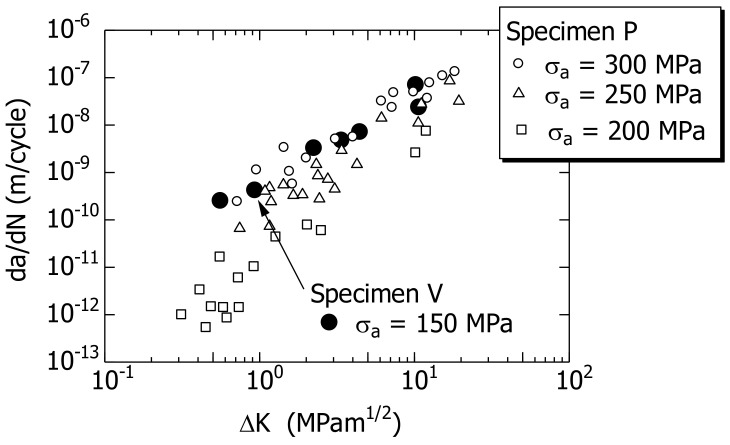
da/dN–ΔK relationships of short surface cracks in specimens P and V.

**Figure 10 materials-14-07538-f010:**
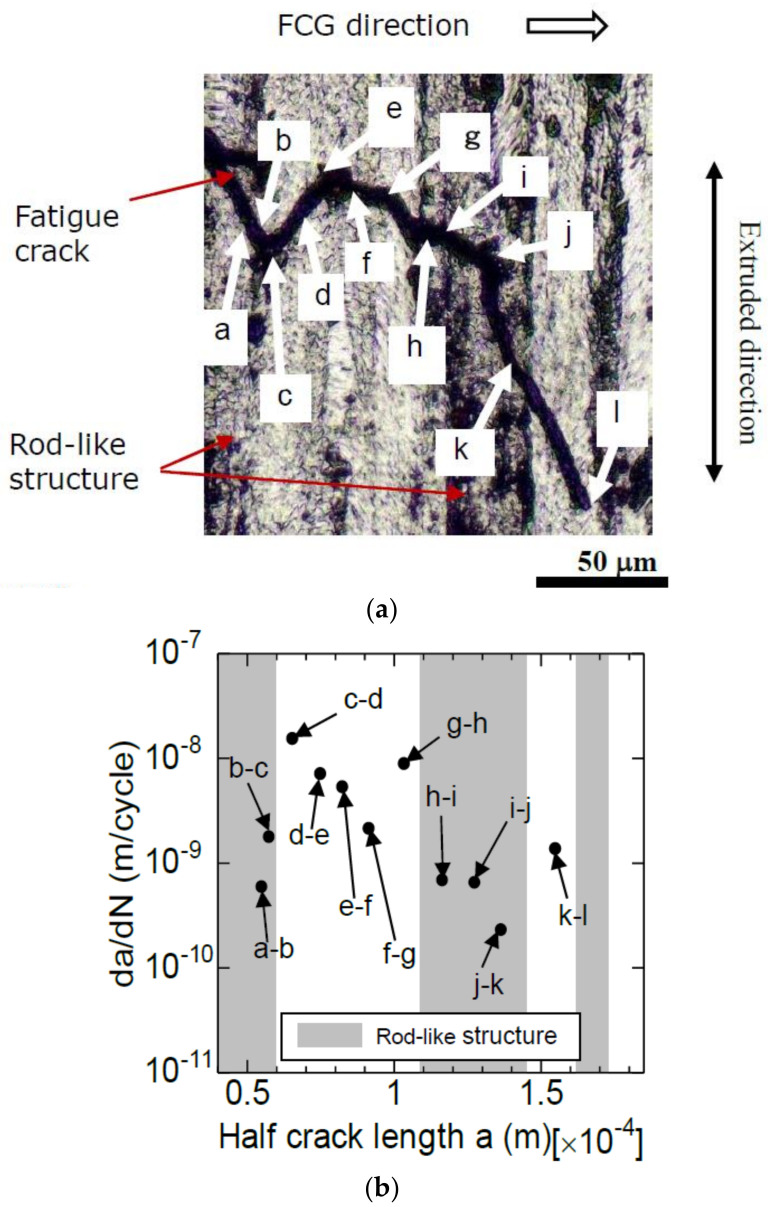
FCG behavior at constant ΔK (2 MPa √m) test (specimen P). (**a**) Rod-like structure; (**b**) Effect of microstructure on local FCG velocity.

**Figure 11 materials-14-07538-f011:**
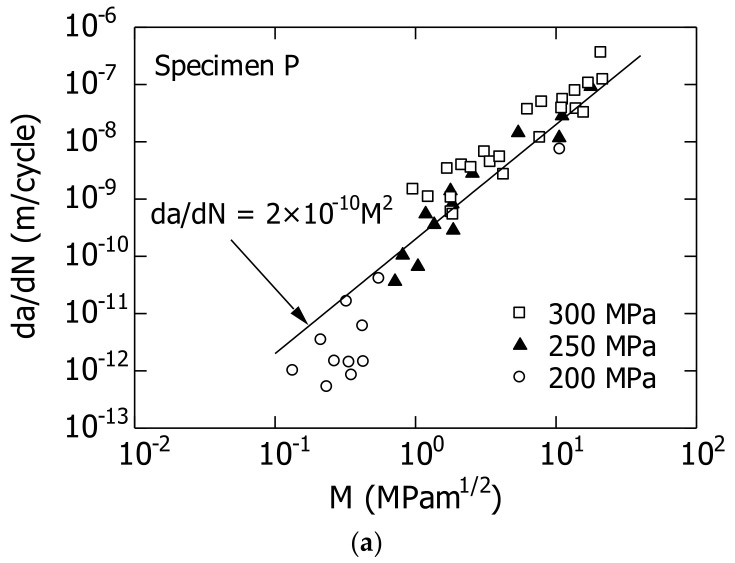
Relationship between da/dN and M of extruded A2024-T3 aluminum alloy. (**a**) Specimen P; (**b**) Specimen V.

**Figure 12 materials-14-07538-f012:**
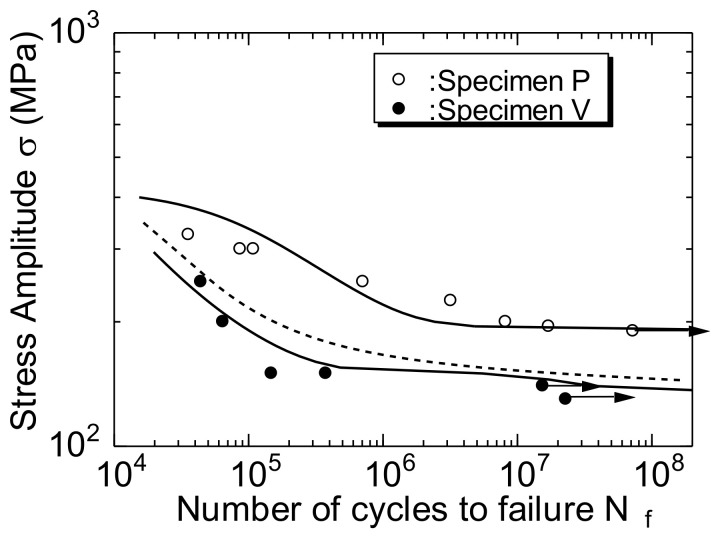
Comparison of S–N curve calculation results and experimental results. The solid lines in the figure are the S–N curves calculated for specimens P and V using Equation (10).

**Figure 13 materials-14-07538-f013:**
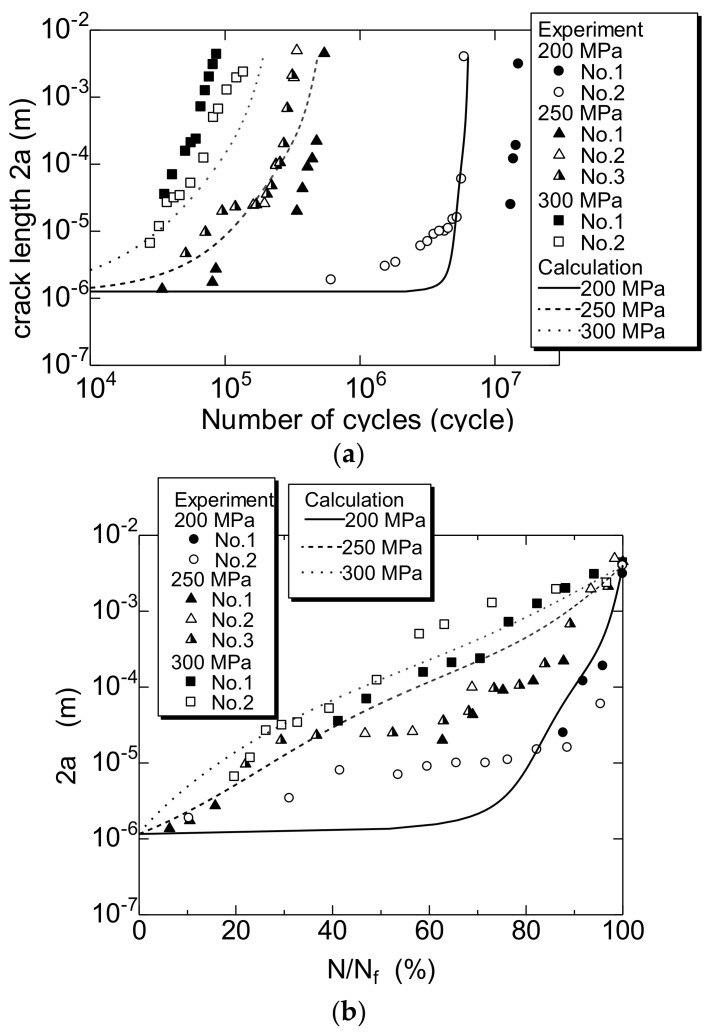
Comparison of the calculation result of the fatigue crack growth curve of the specimen P and the experimental result. (**a**) 2a–N; (**b**) 2a–N/N_f_.

**Figure 14 materials-14-07538-f014:**
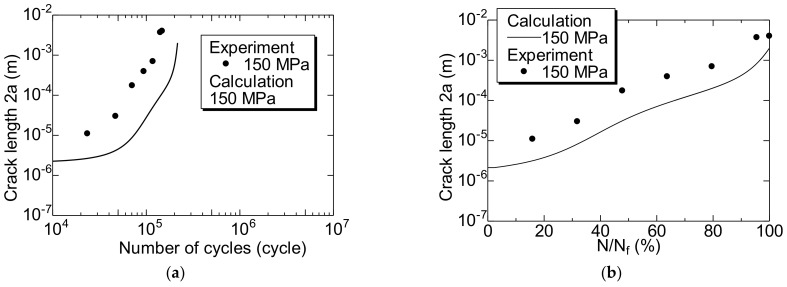
Comparison of the calculation result of the fatigue crack growth curve of the specimen V and the experimental result. (**a**) 2a–N; (**b**) 2a–N/N_f_. The solid lines in these figures are the calculated results using Equation (11).

**Table 1 materials-14-07538-t001:** Chemical composition of A2024-T3 (wt.%).

Cu	Mg	Mn	Zn	Fe	Al
4.25	1.32	0.56	0.25	0.12	Bal.

**Table 2 materials-14-07538-t002:** Mechanical properties of A2024-T3 (Specimen P).

Yield Strength	Tensile Strength	Elongation
420 MPa	570 MPa	17.0%

**Table 3 materials-14-07538-t003:** Results of microscopic tissue identification and microhardness values.

Location	Phase A (Rod-Like Structure)	Phase B (Black Area within A Phase)	Phase C	Phase D (Inclusion within C Phase)
Identification of microscopic tissue	Al–Cu–Mn intermetallic compound	Local segregation of Al–Cu–Mn intermetallic compounds	Al matrix	Cu–Mg-based intermetallic compound
Hardness, HV	182	207	142	213

**Table 4 materials-14-07538-t004:** Parameter values used to calculate M.

Specimen	ΔK_opmax_ MPa$\sqrt{m}$	ΔK_effth_ MPa$\sqrt{m}$	k m^−1^	r_e_ μm
P	4	1.4	13,000	0.58
V	4	1.4	13,000	1.02

## Data Availability

The data presented in this study are available on request from the corresponding author.

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
