# Peer review of "Effect of Rod-like Structure on Fatigue Life, Short Surface Crack Initiation and Growth Characteristics of Extruded Aluminum Alloy A2024 (Analysis via Modified Linear Elastic Fracture Mechanics)"

_materials, 2021, doi:10.3390/ma14247538_

Round 1

Reviewer 1 Report

Manuscript number: materials-1483743 - Effect of rod-like structure on fatigue life, short surface crack initiation, and growth characteristics of extruded aluminum alloy A2024

This is a very well-prepared manuscript of excellent relevance in the field of materials fatigue. The authors studied the effects of extrusion direction on the fatigue strength of A2024 aluminum alloy, focusing on crack initiation and propagation under rotating bending. However, I will provide some suggestions for improvement and questions that will help the reader to better understand the work.

  • Pag 1 line 32 – “If a structural member is used for a long duration…” this statement is very vague and inaccurate, the phenomenon of fatigue can occur in short periods of time and results from cyclic stress, I suggest that this paragraph be revised.
  • Pag1 line 37 – “FCG” as this is the first time it appears in the text, the authors must describe its meaning in full.
  • Page 1 – line 39 “… 20kHz” authors should refer to the fatigue testing regimen, i.e., very high cycle fatigue.
  • Page 2 – line 92 “Table 1. Chemical composition (wt.%).” Authors must include "A 2024-T3" in the caption of Table 1.
  • Page 3 – line 94 “Table 2. Mechanical Properties (Specimen P). Authors must include "A 2024-T3" in the caption of Table 1.
  • Page 3, for Figure 1, sub-figure (a) should be broken down into (a) and (b), also in the caption of Figure 1, each sub-figure (a)... (b) ... (c)... should be described.
  • Page 5, Figure 2 caption, each sub-figure (a)... (b) should be described.
  • Page 6, figure 3, sub-figure b, left side, should be enlarged to better understand the geometry of the specimen and its dimensions (the length of the V specimen should be indicated).
  • For sample V, was the effect of welding on the fatigue strength of the material considered? Describe the welding process used (thermal stress from welding may affect the thermal treatment of material T3). Was the size effect also taken into account in the correlation between the two specimens?
  • Page 7, line 249, a semicircular surface crack is assumed. What is the meaning of this assumption? By experimentation it will be possible to test the truth of this assumption by inspection of the surface crack, it is important to clarify this issue.
  • On page 7, line 249, a constant value for Y is given, is this correct? This geometric factor Y determined taking into account the load type? (Rotating bending)
  • Page 7, line 249, it is mentioned that a kt is used, but kt is not valid for LEFM, why do the authors use this correction?
  • Page 7, figure 4, the calculated lines should be removed, at this point, their meaning cannot be understood, I suggest placing this figure with these lines in section 5.2. Also, in this figure, the arrows must be within the diagram and their meaning explained in the manuscript text.
  • On page 10, figure 7 (b), "crack length" is missing in the description of the vertical axis, also sub-figures (a) and (b) need to be described in the caption.
  • Page 11, Figure 8, subfigures (a) and (b) must be described in the figure caption.
  • Page 13, Figure 10 (a), based on crack propagation and loading direction, it appears that crack propagation is determined by KI, KII, and KIII, in which way does equation (1) account for this combined effect?
  • Page 15, Table 4, the units should be in the first row
  • Page 16, line 550, The authors mention Figure (b), please add the respective number
  • Page 16, line 579, the presentation of the calculated results should be close to the respective formulas. I suggest removing the calculated results from figures 7 and 8 and creating new figures with the calculated data and placing them near this paragraph.

Author Response

To Professor:

We would like to express our sincere gratitude to you for scrutinizing this paper and providing valuable opinions regarding its improvement. We have improved the paper according to your comments. We would appreciate it if you allowed for our paper to be published in Materials.

                        Dr. Kenichi Masuda

Reviewer 1

Comments and Suggestions for Authors

This is a very well-prepared manuscript of excellent relevance in the field of materials fatigue. The authors studied the effects of extrusion direction on the fatigue strength of A2024 aluminum alloy, focusing on crack initiation and propagation under rotating bending. However, I will provide some suggestions for improvement and questions that will help the reader to better understand the work.

(1) Pag 1 line 32 – “If a structural member is used for a long duration…” this statement is very vague and inaccurate, the phenomenon of fatigue can occur in short periods of time and results from cyclic stress, I suggest that this paragraph be revised.

[Answer]

The text has been revised as follows.

Page 1, line 32:

“If a structural member is repeatedly stressed, cracks may appear from the deteriorated section, which could eventually result in its destruction.”

(2)Pag1 line 37 – “FCG” as this is the first time it appears in the text, the authors must describe its meaning in full.

[Answer]

The following explanation has been added to the text.

Page 1, line 37:

“The fatigue crack growth (hereinafter referred to as FCG) properties of Al alloys---"

(3) Page 1 – line 39 “… 20kHz” authors should refer to the fatigue testing regimen, i.e., very high cycle fatigue.

[Answer]

The following explanation has been added to the text.

Page 1, line 39:

“--- to study the FCG behavior in a very high cycle fatigue region --”

(4) Page 2 – line 92 “Table 1. Chemical composition (wt.%).” Authors must include "A 2024-T3" in the caption of Table 1.

[Answer]

 The following description has been added to the captions in Table 1.

“Table 1. Chemical composition of A2024-T3 (wt.%).”

(5) Page 3 – line 94 “Table 2. Mechanical Properties (Specimen P). Authors must include "A 2024-T3" in the caption of Table 1.

[Answer]

 The following description has been added to the captions in Table 2.

 “Table 2. Mechanical properties of A2024-T3.”

(6) Page 3, for Figure 1, sub-figure (a) should be broken down into (a) and (b), also in the caption of Figure 1, each sub-figure (a)... (b) ... (c)... should be described.

[Answer]

Figure 1.:

According to your comments, Figure 1 now consists of sub-figures (a), (b), and (c). In addition, an explanation has been added to the caption for Figure 1.

(7) Page 5, Figure 2 caption, each sub-figure (a)... (b) should be described.

[Answer]

Figure 2.:

A description for each sub-figure has been added to the caption for Figure 2.

(8) Page 6, figure 3, sub-figure b, left side, should be enlarged to better understand the geometry of the specimen and its dimensions (the length of the V specimen should be indicated).

[Answer]

Figure 3.:

Sub-figure (b) is now enlarged, and the length of the V specimen is shown.

(9) For sample V, was the effect of welding on the fatigue strength of the material considered? Describe the welding process used (thermal stress from welding may affect the thermal treatment of material T3). Was the size effect also taken into account in the correlation between the two specimens?

[Answer]

Regarding the welding process:

Before welding, an adhesive was used to bond the test piece to the grip portion. Subsequently, spot welding was performed at four locations so that the effect of welding on the specimen did not increase.

Regarding the size effect:

Two specimens of the same material but different shapes (specimens P and V) were prepared, and their S-N curves were compared. It was confirmed from the S-N curves that there was no significant difference between the two specimens.

The above explanation has been added to page 6, lines 216-222.

(10) Page 7, line 249, a semicircular surface crack is assumed. What is the meaning of this assumption? By experimentation it will be possible to test the truth of this assumption by inspection of the surface crack, it is important to clarify this issue.

[Answer]

The fatigue fracture surface of the specimen was observed at a stress amplitude of 250 MPa using SEM. A semi-circular surface crack with a length of 6.7 μm and a depth of 3.5 μm was observed at the origin of the fracture. It was also observed that the shape of the crack could be approximated by a semicircle up to a length of approximately 2 mm.

The above explanation has been added to the text.

page 7, lines 260-264:

“The fatigue fracture surface of the specimen was observed at a stress amplitude of 250 MPa using SEM. A semi-circular surface crack with a length of 6.7 μm and a depth of 3.5 μm was observed at the origin of the fracture. It was also observed that the shape of the crack could be approximated by a semicircle up to a length of approximately 2 mm."

(11) On page 7, line 249, a constant value for Y is given, is this correct? This geometric factor Y determined taking into account the load type? (Rotating bending)

[Answer]

  As mentioned in answer (10), the shape of the surface crack can be approximated by a semicircle up to a length of approximately 2 mm. It is considered that when the length of the crack is short, the influence of the stress gradient due to bending is unlikely to occur. The description has been added to the text.

(12)Page 7, line 249, it is mentioned that a kt is used, but kt is not valid for LEFM, why do the authors use this correction?

[Answer]

As shown in Figure 3, the specimen does not have a parallel section and has an arcuate notch. Therefore, the stress at the minimum diameter of the specimen was determined using the formula for the strength of materials. The computed value was multiplied by the stress concentration coefficient of 1.04 due to the notch in the arc to obtain the stress amplitude σa.

The following description has been added to the text.

page 7, lines 266-270:

“As shown in Fig. 3, the specimen does not have a parallel section and has an arcuate notch. Therefore, the stress at the minimum diameter of the specimen was determined using the formula for the strength of materials. The computed value was multiplied by the stress concentration factor of 1.04 due to the notch in the arc to obtain the stress amplitude σa.”

(13) Page 7, figure 4, the calculated lines should be removed, at this point, their meaning cannot be understood, I suggest placing this figure with these lines in section 5.2. Also, in this figure, the arrows must be within the diagram and their meaning explained in the manuscript text.

[Answer]

The calculated results in Fig. 4 have been deleted. A new Figure 13 was created in Section 5.2 to compare the experimental results with the calculated results. The text has been changed to accommodate the changes. In addition, a description of the arrows in Figure 4 has been added.

(14) On page 10, figure 7 (b), "crack length" is missing in the description of the vertical axis, also sub-figures (a) and (b) need to be described in the caption.

[Answer]

"Crack length" was added to the title of the vertical axis in Fig. 7 (b). Sub-figure descriptions were also added to the caption in Figure 7.

(15) Page 11, Figure 8, subfigures (a) and (b) must be described in the figure caption.

[Answer]

Sub-figure descriptions have been added to the caption for Figure 8.

(16) Page 13, Figure 10 (a), based on crack propagation and loading direction, it appears that crack propagation is determined by KI, KII, and KIII, in which way does equation (1) account for this combined effect?

[Answer]

As you have pointed out, the crack does not grow in the direction perpendicular to the loading direction locally. Therefore, only the Mode-I component of fatigue cracks was considered. Finally, the crack was projected onto a plane orthogonal to the load direction to measure its length.

The above explanation has been added to the text.

page 13, lines 411-414:

The crack did not grow in the direction perpendicular to the load direction (extrusion direction); instead, they grew locally in the diagonal direction. Therefore, the crack was projected on a plane orthogonal to the load direction in order to measure its length.

(17) Page 15, Table 4, the units should be in the first row.

[Answer]

Table 4 has been changed according to your suggestions.

(18) Page 16, line 550, The authors mention Figure (b), please add the respective number

[Answer]

The part you have pointed out has been added.

(19) Page 16, line 579, the presentation of the calculated results should be close to the respective formulas. I suggest removing the calculated results from figures 7 and 8 and creating new figures with the calculated data and placing them near this paragraph.

[Answer]

The calculation results in Figures 7 and 8 have been deleted. In Section 5.2, new Figures 14 and 15 have been created to compare the experimental and calculated results. The text has been changed to accommodate the changes.

Reviewer 2 Report

The authors intended to clarify the effects of rod-like structures on fatigue crack initiation and growth behavior in A2024-T3 extruded Al alloys. Before making a decision, the authors are strongly required to organize the paper in order and follow the basic structure in terms of introduction, materials and methods, results, discussion and conclusions. Moreover, the language should be concise and polished. Several additional comments are shown. 1)  What's the meaning of FCG? 2) The scattered figures should be re-arranged.  3) About the novelty, I don't understand it. The different direction shows different effects on the fatigue life, which is clearly understood. The fatigue life might be related to the perpendicular direction because it is cracked first. In my opinion, the authors should make effects to produce the rod-like structure in the perpendicular direction, which might be more interesting.

Author Response

To Professor:

We would like to express our sincere gratitude to you for scrutinizing this paper and providing valuable opinions regarding its improvement. We have improved the paper according to your comments. We would appreciate it if you allowed for our paper to be published in Materials.

                        Dr. Kenichi Masuda

Reviewer 2:

The authors intended to clarify the effects of rod-like structures on fatigue crack initiation and growth behavior in A2024-T3 extruded Al alloys.

Before making a decision, the authors are strongly required (1) to organize the paper in order and follow the basic structure in terms of introduction, materials and methods, results, discussion and conclusions.

Moreover, (2) the language should be concise and polished.

Several additional comments are shown.

(3) What's the meaning of FCG?

(4) The scattered figures should be re-arranged. 

(5) About the novelty, I don't understand it. The different direction shows different effects on the fatigue life, which is clearly understood. The fatigue life might be related to the perpendicular direction because it is cracked first. In my opinion, the authors should make effects to produce the rod-like structure in the perpendicular direction, which might be more interesting.

(1) --to organize the paper in order and follow the basic structure in terms of introduction, materials and methods, results, discussion and conclusions.---

 [Answer]

It was confirmed that the paper conforms to the basic structure described above.

(2) ---the language should be concise and polished.

[Answer]

The paper was reviewed again and the necessary changes were made.

(3) What's the meaning of FCG?

[Answer]

“FCG” is an abbreviation for “fatigue crack growth” and the explanation has been added to the text.

(4) The scattered figures should be re-arranged.

[Answer]

The arrangement of the figures was improved so that they are more compact throughout the paper.

(5) About the novelty, -----

[Answer]

The novelty of this research can be summarized by the following three points.

* It was revealed that a rod-like microstructure was present in the A2024-T3 extruded material. The mechanism by which it affects fatigue life and crack growth behavior has been elucidated, and the effect of the identified rod-like microstructure has been quantitatively elucidated.

* A quantitative evaluation of fatigue life and FCG behavior was performed using modified linear fracture mechanics, and it was shown to be consistent with the experimental results.

*By combining the replica method with a scanning atomic force microscope, fatigue cracks with a length of several microns were continuously observed, and it was determined that fatigue cracks occurred in the early stages of the fatigue process.

 This result has been added to the conclusion.

Conclusions:

 (6) By combining the replica method with a scanning atomic force microscope, fatigue cracks with a length of several microns were continuously observed, and it was determined that fatigue cracks occurred in the early stages of the fatigue process.

For future research, we intend to investigate a method for generating rod-like structures.

Round 2

Reviewer 1 Report

The article has been substantially improved. In my opinion, the article meets the necessary conditions for its publication in Materials.

Reviewer 2 Report

The authors did make some improvements in the revised manuscript and it was worth publishing on your journal.